# Characterization and quantitative trait locus mapping of late-flowering from a Thai soybean cultivar introduced into a photoperiod-insensitive genetic background

Fei Sun[1], Meilan Xu[1,2], Cheolwoo Park[1], Maria Stefanie Dwiyanti[1], Atsushi J. Nagano[3], Jianghui Zhu[1], Satoshi Watanabe[4], Fanjiang Kong[5], Baohui Liu[5], Tetsuya Yamada[1], Jun Abe[1]*

**1** Research Faculty of Agriculture, Hokkaido University, Sapporo, Hokkaido, Japan, **2** Key Laboratory of Soybean Molecular Design Breeding, Northeast Institute of Geography and Agroecology, Chinese Academy of Sciences, Harbin, China, **3** Faculty of Agriculture, Ryukoku University, Otsu, Shiga, Japan, **4** Faculty of Agriculture, Saga University, Saga, Saga, Japan, **5** School of Life Sciences, Guangzhou University, Guangzhou, China

* jabe@res.agr.hokudai.ac.jp

**Data Availability Statement:** All relevant data are within the paper and its Supporting Information files.

## Abstract

The timing of both flowering and maturation determine crop adaptability and productivity. Soybean (*Glycine max*) is cultivated across a wide range of latitudes. The molecular-genetic mechanisms for flowering in soybean have been determined for photoperiodic responses to long days (LDs), but remain only partially determined for the delay of flowering under short-day conditions, an adaptive trait of cultivars grown in lower latitudes. Here, we characterized the late-flowering (LF) habit introduced from the Thai cultivar K3 into a photoperiod-insensitive genetic background under different photo-thermal conditions, and we analyzed the genetic basis using quantitative trait locus (QTL) mapping. The LF habit resulted from a basic difference in the floral induction activity and from the suppression of flowering, which was caused by red light-enriched LD lengths and higher temperatures, during which *FLOWERING LOCUS T* (*FT*) orthologs, *FT2a* and *FT5a*, were strongly down-regulated. QTL mapping using gene-specific markers for flowering genes *E2*, *FT2a* and *FT5a* and 829 single nucleotide polymorphisms obtained from restriction-site associated DNA sequencing detected three QTLs controlling the LF habit. Of these, a QTL harboring *FT2a* exhibited large and stable effects under all the conditions tested. A resequencing analysis detected a nonsynonymous substitution in exon 4 of *FT2a* from K3, which converted the glycine conserved in FT-like proteins to the aspartic acid conserved in TERMINAL FLOWER 1-like proteins (floral repressors), suggesting a functional depression in the FT2a protein from K3. The effects of the remaining two QTLs, likely corresponding to *E2* and *FT5a*, were environment dependent. Thus, the LF habit from K3 may be caused by the functional depression of FT2a and the down-regulation of two *FT* genes by red light-enriched LD conditions and high temperatures.

**Funding:** This work was supported in part by Grants-in-Aid for Scientific Research from the Ministry of Education, Culture, Sports, Science, and Technology of Japan (17K07579) to J Abe; by the National Natural Science Foundation of China (grant nos. 31501330) to M Xu, the Program of the General Program of the National Natural Science Foundation of China (grant nos. 31771815) to B Liu; by the National Natural Science Foundation of China (grant nos. 31725021) to F Kong; and by the Natural Key R&D Program of China (2017YFE0111000 and 2016YFD0100400) to F Kong.

**Competing interests:** The authors have declared that no competing interests exist.

## Introduction

The photo-thermal regulation of flowering and maturation determines the adaptability and productivity of crops. Diverse genotypic combinations at several flowering loci enable crops to maximize their productivity under various environmental conditions. Soybean [*Glycine max* (L.) Merr.] is a crop that is cultivated over a wide range of latitudes. This wide adaptability occurs owing to natural variations at a number of major genes and quantitative trait loci (QTLs) that control the timing of both flowering and maturation [1, 2]. Recent genome-wide and flowering gene-specific association analyses have detected additional DNA polymorphisms in the orthologs of *Arabidopsis* flowering genes that are associated with variations in soybean flowering time [3–9].

Molecular and genetic mechanisms of photoperiod responses related to flowering, particularly to long days (LDs), an adaptive trait of soybean cultivars grown in high latitudes, have been gradually but steadily disclosed. Flowering in soybean is controlled by three partly-inter-related modules; phytochrome A (PHYA)-E1, GIGANTEA (GI)-CONSTANS (CO) and miRNA-dependent modules [2]. Of these, the PHYA-E1 module is the main regulator in the photoperiodic flowering of soybean as demonstrated by night-break responses [10]. The *E1* gene and its homologs, *E1La* and *E1Lb*, are legume-specific putative transcription factors that repress the transcription of soybean *FLOWERING LOCUS T* (*FT*) orthologs, *FT2a* and *FT5a*, during LDs under the control of phytochrome A proteins encoded by *E3* and *E4* [10–14]. *E2* is a soybean ortholog of *Arabidopsis GI*, which inhibits flowering under LD conditions through a pathway distinct from the PHYA-regulated E1 pathway [10, 15]. Allelic combinations at these flowering loci produce diverse flowering phenological events under different environmental conditions worldwide [16–24]. In particular, a reduced or lack of sensitivity to LD lengths is conferred by loss-of-function alleles at the *E1*, *E1Lb*, *E3* and *E4* loci, as well as a transcriptionally upregulated *FT5a* allele [25–27]. In addition to these early-flowering alleles, at the *E9* locus, which encodes FT2a, the recessive allele *e9*, having reduced *FT2a* transcription, contributes to the retention of a sufficient vegetative growth duration by negatively acting on flowering, particularly in early-flowering plants homozygous at the *E1* locus for hypomorphic or loss-of-function alleles (*e1-as* and *e1-nl*, respectively) [28]. With various combinations of genes acting positively or negatively on flowering, soybean cultivars may fine-tune the timing of flowering and maturation to produce higher yields under LD conditions and the limited frost-free season found at high latitudes.

In contrast to the lack of, or reduced, photoperiod sensitivity to LDs that has enabled soybean cultivars to adapt to high latitudes, a repression of flowering under short day (SD) conditions, the inductive phase of flowering, is an indispensable trait for soybean cultivars grown at lower latitudes having hot temperatures. The delayed flowering under SD conditions has been referred as long juvenility in soybean. The juvenile phase is an early growing phase insensitive to photoperiods; therefore, soybean cultivars with long juvenile (LJ) periods can produce greater seed yields, compared with those having normal juvenile periods, by retaining sufficient vegetative growth levels under SD conditions [29]. The LJ period appears to be controlled by a number of recessive genes (reviewed in [29]), including two major genes *j* [30] and *e6* [31]. The *j* gene is a loss-of-function allele at the *J* locus that encodes the *Arabidopsis* EARLY FLOWERING 3 ortholog [32, 33]. A mapping study further demonstrated that the *E6* locus is tightly linked to the *J* locus in chromosome (Chr) 04 and that the *e6* allele is a recessive allele at the *J* locus or its tightly linked gene [34].

QTL mapping studies have also revealed the involvement of novel *LJ* genes, other than *j* and *e6*, in delayed flowering under SD conditions [16, 22, 34]. Liu et al. [16] identified QTLs for the variations of flowering time that segregated in different latitudinal environments in a

cross between early and late flowering Korean soybean cultivars. Of these, two QTLs located in Chr4 and 16 were responsible for the control of flowering under the SD conditions found at lower latitudes. Lu et al. [22] reported that two maturity loci, *E2* and *E3*, and two novel QTLs located in Chr16 that were involved in the variations of flowering time that segregated under SD conditions in a cross between photoperiod-insensitive cultivar AGS292 and the Thai LJ cultivar K3. Furthermore, the QTL mapping in a cross between the LJ cultivars Paranagoiana and PI159925 revealed two QTLs that regulated flowering time under SD conditions, one corresponding to *E6/J*, or a linked novel locus, and one located in Chr2 (D1b linkage group) [34].

In this study, we characterized the LF habit from the Thai cultivar K3 in a photoperiod-insensitive genetic background and studied the genetic basis using QTL mapping. In this manner, the effects of the PHYA-E1 module on the photoperiodic flowering were removed. Here, we report that the LF habit may require a low floral induction activity and the suppression of flowering caused by red light-enriched LD and hot temperatures. Three QTLs likely corresponding to *E2* and two *FT* orthologs, *FT2a* and *FT5a*, may be involved in controlling the LF habit in the photoperiod-insensitive genetic background.

## Materials and methods

### Plant materials

AGS292 and the RIL-#16 (AK16) selected from the recombinant inbred line (RIL) population developed from a cross between AGS292 and the Thai late-flowering cultivar K3 [22, 35] were used in this study. The maturity genotypes at *E1–4* and *E9* were *E1/e2/e3/e4/E9* in AGS292 and *E1/E2/e3/e4/E9* in AK16. Both lines possessed the dominant allele at the *J* locus (*J/J*). AK16 was the latest flowering line of 16 photoperiod-insensitive RILs having the *e3/e4* genotype [22]. It possessed alleles from K3 in 120 (44%) of 276 simple sequence repeat (SSR) or insertions and deletions (indels) markers used for mapping. The genomic region from K3 covered the QTLs responsible for the variations in flowering time under SD conditions in linkage groups (LG) J (chromosome 16; Chr16) and O (Chr10), harboring *FT2a*, *FT5a* and *E2* [22]. In total, 75 $F_6$ families developed using the single seed descent method from $F_2$ plants of a cross between the two lines were used for the QTL mapping.

### Growing conditions

AGS292 and AK16 were cultivated under different photoperiod and thermal conditions. Four day-length conditions of 8, 12, 16 and 20 h were set in a greenhouse in the winters of 2016 to 2018. High intensity discharge lamps (HONDA-T, Panasonic Co, Osaka, Japan) were used during the daytime. The average photosynthetically active photon flux density was 120 μmol $m^{-2}$ $s^{-1}$, and the red-to-far red (R:FR) ratio was 4.5 at 1 m below the light source. Air temperatures in the greenhouse were adjusted to 25°C, with fluctuations from a minimum of 20°C to a maximum of 28°C. Three consistent temperature conditions of 18°C, 25°C and 32°C and a varying temperature condition of 32°C in the daytime and 25°C in the nighttime were set in the growth chambers. Lighting was supplied for 16 h using a combination of fluorescent and incandescent lamps with an average photosynthetically active photon flux density of 150 μmol $m^{-2}$ $s^{-1}$ and an R:FR ratio of 7.0. For the genetic analysis, $F_6$ families and parents were cultivated under SD (12 h) and LD (20 h) conditions in the same photo-thermal setting in a greenhouse in the winter of 2017. The $F_6$ families and parents were also cultivated under natural day (ND) conditions and incandescent-induced LD (ILD) conditions in an experimental farm of Hokkaido University, Sapporo (43°07′N, 141°35′E) in 2017. The ILD condition was generated by extending ND to 20 h using supplemental lighting of incandescent lamps with an R:FR ratio of 0.7 from 2:00 to 7:00 and 18:00 to 22:00 every day from sowing to August 10th.

## Cultivation methods

In the controlled experiments in the greenhouse and growth chambers, seeds were directly sown into plastic pots (15 cm in diameter and depth), and then, thinned to four plants per pot. In the field experiments, seeds were sown in paper pots (Paperpots No.2, Nippon Beet Sugar Manufacturing Co., Tokyo, Japan) on 28 May 2017 and put outdoors under ND or ILD conditions until transplanted into the field. The daily mean outdoor air temperature from the sowing date to the end of July was 18.9˚C, with a minimum of 9.6˚C and a maximum of 27.3˚C. Flowering times were recorded individually and expressed as days after sowing (DAS).

## DNA extraction and marker analysis

Total DNA was extracted by bulk from young leaves of four plants per family using the modified CTAB method [36]. Cleaved amplified polymorphic sequence markers were developed to detect SNPs in the fourth exon of *FT2a* and the 3′ untranslated region (UTR) of *FT5a*. The products amplified by PCR using region-specific primers were digested with the restriction enzyme HinfI (*FT2a*) or PvuII (*FT5a*), and were separated by electrophoresis in a 2.5% agarose gel, stained with ethidium bromide and visualized under UV light. The genotypes at the *E2* locus were determined using the functional DNA marker developed by Tsubokura et al. [20]. The primers used are listed in S1 Table.

## Restriction site-associated DNA sequence analysis

Total DNA was digested using the restriction enzymes BglII and EcoRI to create a DNA library for double-digest restriction site-associated DNA sequencing (ddRAD-Seq) [37, 38]. Sequencing was performed with 51-bp single-end reads in one lane of a HiSeq2000 Sequencer (Illumina, San Diego, CA, USA) by Macrogen (Seoul, South Korea). The resulting reads were trimmed with Trimmomatic ver 0.3 [39] using the following parameters: LEADING:19, TRAILING:19, SLIDINGWINDOW:30:20, AVGQUAL:20, and MINLEN:51. These ddRAD-Seq analyses were carried out by Clockmics, Inc. (Izumi, Osaka, Japan). The trimmed reads were mapped to the soybean reference genome Williams 82.v2 using Bowtie2 [40] with the default parameter settings. SNP calling was performed using the Genome Analysis Toolkit (GATK) Unified Genotyper [41]. The imputation of missing genotypes in RILs based on the parental SNP data was performed using Beagle 4.0 [42]. Filtering for monomorphic SNPs and SNPs having many missing calls was performed using TASSEL.5.2.31 [43] with the following parameters: a minimum call rate per SNP of 90% and a minimum allele frequency of 0.05 (to remove monomorphic SNPs). Using a custom script in R (https://www.R-project.org/), the nucleotide information was converted to the AB genotype with parents A and B being AGS292 and AK16, respectively. All the heterozygotes were converted as missing genotypes. Further filtering for duplicated markers or markers having switch alleles was performed in R/QTL [44], and resulted in a final set of 829 SNPs.

## QTL mapping

QTL IciMapping ver 4.1 [45] was used to construct a linkage map with three gene-specific and 829 SNP markers. The input algorithm, which re-estimated recombination frequencies and genetic distances without changing the marker order in the input file, was used to determine the order of the markers on the genetic map. The sum of the adjacent recombinant frequencies with a window size of 5 was used as a rippling criterion for fine tuning the markers. Recombination frequencies between linked loci were transformed into centimorgan (cM) distances using Kosambi's mapping function. The inclusive composite interval mapping of additive

QTLs implemented in QTL IciMapping version 4.1 [45] was used to detect the QTLs at a 5% level after 1,000 genome-wide permutation tests.

## Construction of *FT2a* and *FT5a* sequences based on whole-genome resequencing data

Raw reads of AGS292 and K3 from next-generation sequencing on Illumina HiSeq XTen were aligned to the soybean reference genome Williams 82.v2 [46]. The alignment was performed using Bowtie2-2.2.9 [47]. The resulting alignment was further processed to remove duplicate reads and to fix mate information using Picard tools (http://broadinstitute.github.io/picard). GATK ver 3.8 [41] was used to realign small indels. Subsequently, variants (SNP and indels) were called using the GATK Unified Genotyper function that filtered out reads having mapped base quality Phred scores of less than 20. Using the reference genome Williams 82.v2 and a SNP dataset for each variety, sequences of *FT2a* and *FT5a* were reconstructed using the FastaAlternateReferenceMaker function available in GATK. A plant *cis*-acting regulatory DNA elements [48] analysis was carried out to detect known *cis*-elements in the *FT2a* and *FT5a* sequences.

## Expression analyses

Expression analyses of *FT2a*, *FT5a* and *E2* were performed using fully expanded new leaves of AGS292 and AK16 grown in the greenhouse and in the growth chambers. Leaves were sampled at Zeitgeber times of 3, 12 and 21 h in two different growth stages, the second and fourth trifoliate-leaf stages. All the samples were immediately frozen in liquid nitrogen and stored at −80˚C. Total RNA was isolated from each sample using TRIzol Reagent (Invitrogen, Waltham, MA, USA). cDNA was synthesized from total RNA (1 μg) using an oligo (dT) 20 primer with M-MLV Reverse Transcriptase (Invitrogen) in a 20-μL volume. The transcript abundance levels of *FT2a*, *FT5a* and *E2* were determined using quantitative real-time PCR (qRT-PCR), as described previously [12]. Briefly, each qRT-PCR mixture (20 μL) contained 0.05 μL of the cDNA synthesis reaction mixture, 5 μL of 1.2 μM primer premix and 10 μL of TB Green Premix ExTaq II (TaKaRa, Kyoto, Japan). A CFX96 Real-Time System (Bio-Rad, Hercules, CA, USA) was used to quantify expression levels. The PCR cycling conditions were 95˚C for 3 min followed by 40 cycles of 95˚C for 10 s, 58˚C for 20 s, 72˚C for 20 s and 78˚C for 2 s. The mRNA for *Actin-2/7* was used for normalization. A reaction mixture without reverse transcriptase was also used as a control to confirm the absence of genomic DNA contamination. The amplification of a single DNA fragment was confirmed by a melting curve analysis and the gel electrophoresis of the PCR products. Averages and standard errors of relative expression levels were calculated for three independently synthesized cDNAs. Primers used in the expression analyses are listed in S1 Table.

## Results

### The *e3/e4* LF line responds to R-enriched LD, but not to FR-enriched LD, conditions

AGS292 and AK16 are both the *e3/e4* genotype that conditions photoperiod insensitivity in soybean. Under outdoor ND conditions (maximum day length of 15.5 h) in Sapporo, AK16 flowered at 61.8 DAS, which was, on average, 22.3 d later than AGS292, which flowered at 39.5 DAS. However, under ILD conditions, they flowered at almost the same times as under ND conditions (at 62.3 DAS for AK16 and at 43.7 DAS for AGS2982). Thus, the flowering times of both lines were not largely influenced by FR-enriched ILD (FR-LD) conditions, which was

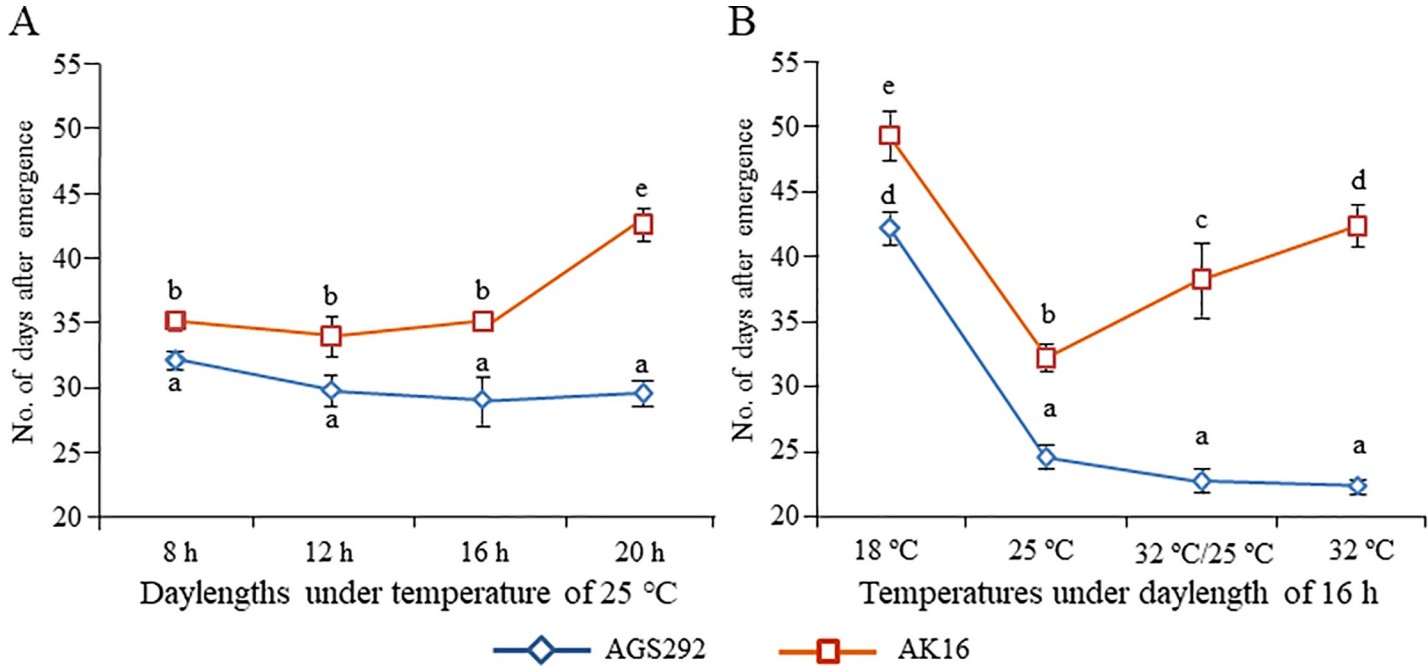

**Fig 1. Flowering times of photoperiod-insensitive soybean lines AGS292 and AK16 under different photo-thermal conditions.** (A) Flowering times under four photoperiod conditions (8, 12, 16 and 20 h) at a constant 25˚C. (B) Flowering times under four thermal conditions, three constant (18˚C, 25˚C and 32˚C) and one variable (32˚C in the daytime and 25˚C in the nighttime) with a 16-h day length. Different letters indicate significant differences at the 5% level in flowering times among lines/conditions as assessed using the Tukey–Kramer method.

consistent with the photoperiod insensitive *e3/e4* genotype. Intriguingly, the two lines responded differently to the R-enriched LD (R-LD) conditions in the greenhouse (Fig 1A). AGS292 flowered at almost the same time (28.9 to 31.8 DAS) under the four different day-length conditions, which ranged from 8 h to 20 h. The flowering of AK16 was delayed by 3.3 d and 5.6 d under 8-h and 16-h day lengths, respectively, with an average of 4.9 d, compared with the flowering time of AGS292. The flowering delay in AK16 increased under R-LD conditions with a 20-h day length; AK16 flowered on average 11.7 d later than AGS292. Thus, AK16 retained the flowering sensitivity under LD conditions produced using high-intensity discharge lamps with greater R:FR ratios, although it did not respond to FR-LD under outdoor conditions.

### Responses of the *e3/e4* LF line to different thermal conditions

We determined the flowering times of AGS292 and AK16 under three thermal conditions, constant 18˚C, 25˚C and 32˚C, at a day length of 16 h using growth chambers (Fig 1B). Flowering time at 25˚C was on average 24.6 and 32.3 DAS in AGS292 and AK16, respectively, with the difference (7.7 d) being significant at the 5% level. The flowering times were delayed at 18˚C by almost the same numbers of days in AGS292 (17.7 d) and AK16 (17.2 d) compared with at 25˚C, indicating that both lines responded similarly to the lower temperature. In contrast, the responses to 32˚C differed between the two lines; AGS292 flowered at 22.4 DAS, which was slightly earlier than at 25˚C (24.6 DAS), while AK16 flowered at 42.5 DAS, which was 20.1 d later than AGS292. The flowering delay was also found in AK16 grown under the varying conditions of 32˚C during the daytime and 25˚C during the nighttime. AGS292 flowered at almost the same time (22.8 DAS) as with the constant 32˚C (22.4 DAS), while AK16 flowered at 38.4 DAS, which was, on average, 15.6 d later than AGS292.

## Segregation of flowering time and QTL analysis

The flowering time trait in $F_6$ families segregated continuously within the ranges of the parental lines under SD (12 h) and R-LD (20 h) conditions in the greenhouse and under ND and FR-LD (20 h) conditions in the experimental farm (Fig 2A–2D). The flowering times of the $F_6$ families correlated positively between the SD and R-LD conditions ($r = 0.537$, $p < 0.01$), and between the ND and FR-LD conditions ($r = 0.537$, $p < 0.01$) (Fig 3A and 3B).

The linkage map covering 868.4 cM was constructed with 829 SNPs obtained from the ddRADseq analysis and three gene-specific markers for *E2*, *FT2a* and *FT5a*. The limited genome coverage resulted from AK16 being a progeny of a cross between AGS292 and K3. It possessed alleles from K3 at only 44% of the marker loci used in the mapping. The composite interval mapping implemented in IciMapping detected three significant QTLs for days to flowering (DTF), *qDTF-10*, *qDTF-16-1* and *qDTF-16-2*, under the four photo-thermal conditions (Table 1, Fig 4). Of these, *qDTF-16-2* was detected under all four conditions, and it solely accounted for 23.9% to 56.1% of the total variation observed. *qDTF-16-1* also exhibited significant allelic effects under all the conditions, except for R-LD. The allelic effects were smaller

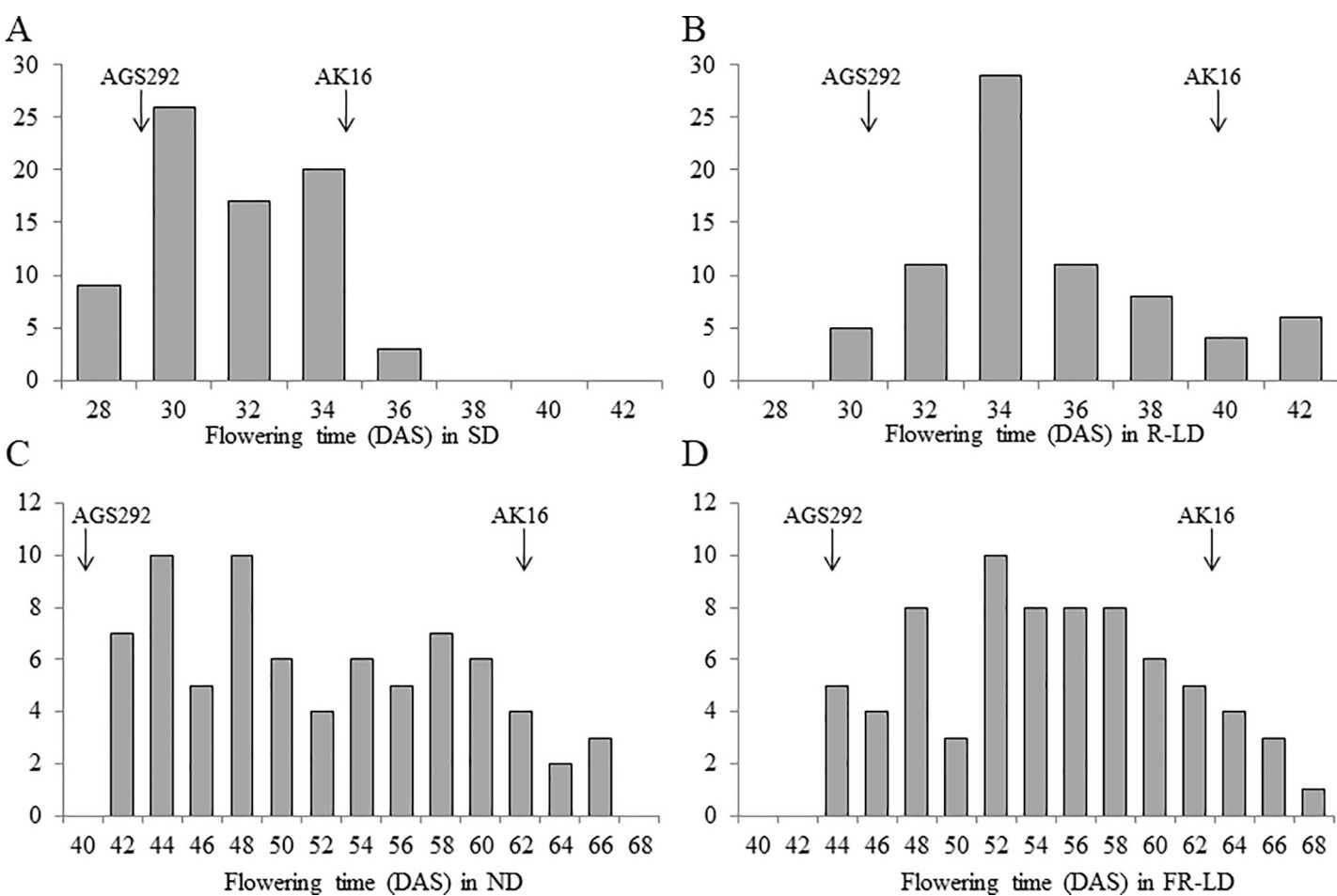

**Fig 2. Frequency distributions of flowering time in the $F_6$ progeny of a cross between soybean lines AGS292 and AK16 under four photo-thermal conditions.** (A) SD conditions (day length of 12 h), (B) R-LD conditions (day length of 20 h) generated with red light-enriched high-intensity discharge lamps, (C) ND conditions in Sapporo (day length of a max. 15.5 h), (D) FR-LD conditions (day length of 20 h) generated with far-red light-enriched incandescent lamps. The SD and R-LD conditions were set in a greenhouse at 25˚C. The ND and FR-LD conditions were set outdoors. The outdoor average temperature from the sowing date to the end of July was 18.9˚C.

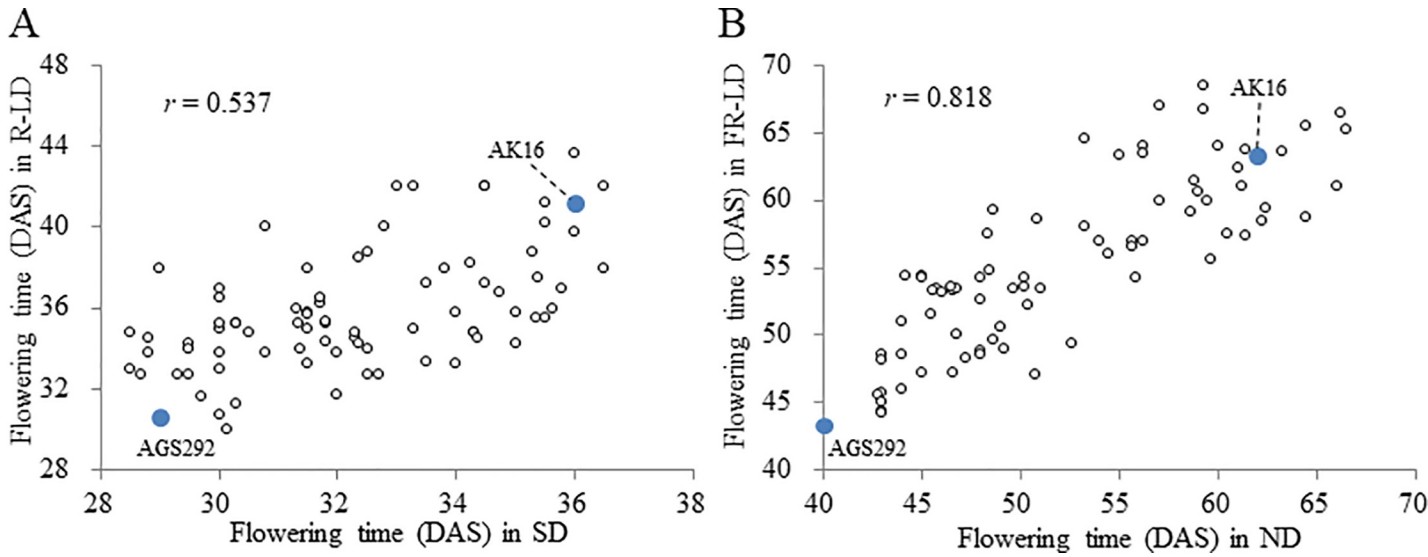

**Fig 3.** Scatter diagrams of flowering times in the F$_6$ progeny of a cross between AGS292 and AK16 (A) between SD and R-LD conditions and (B) between ND and FR-LD conditions.

than those of *qDTF-16-2* under the SD, R-LD and ND conditions, but greater under the FR-LD conditions (Table 1). *qDTF-10* was detected only under the outdoor ND and FR-LD conditions, although the effects were not significant in the latter. Collectively, the detected QTLs accounted for 31.2% under R-LD conditions (40.2% if non-significant *qDTF-16-1* was used) to 72.9% under ND conditions of the total variation observed. The *FT2a* and *E2* gene-specific markers had the greatest logarithm of odds (LOD) scores for *qDTF-16-2* and *qDTF-10*, respectively. In *qDTF-16-1*, the greatest LOD scores occurred in the interval of the *FT5a* gene-specific marker and SNP_4463558 under SD, R-LD and ND conditions, but occurred in the interval of the next two SNPs, SNP_4900158 and SNP_5373087, under FR-LD conditions. The intervals between the left and right markers contained 30, 55 and 167 annotated genes for *qDTF-16-1*, *qDTF-16-2* and *qDTF-10*, respectively, in the reference genome Williams 82.v2 (S2 Table).

**Table 1. Quantitative trait loci for flowering time in a cross between AGS292 and AK16.**

| Environment | QTL | Chr | Left marker | Right marker | LOD | PVE | Additive effect |
|---|---|---|---|---|---|---|---|
| SD (12h) | *qDTF-16-1* | 16 | FT5a (4136378) | SNP_4463558 | 5.1 | 12.3 | 0.7 |
| | *qDTF-16-2* | 16 | FT2a (31114633) | SNP_31642505 | 16.0 | 56.1 | 1.5 |
| R-LD (20h) | *qDTF-16-1* | 16 | FT5a (4136378) | SNP_4463558 | 2.2 [ns] | 9.0 | 0.9 |
| | *qDTF-16-2* | 16 | FT2a (31114633) | SNP_31642505 | 4.6 | 31.2 | 1.5 |
| ND | *qDTF-10* | 10 | E2 (45310798) | SNP_46678320 | 5.6 | 16.0 | 2.6 |
| | *qDTF-16-1* | 16 | FT5a (4136378) | SNP_4463558 | 8.0 | 23.9 | 3.3 |
| | *qDTF-16-2* | 16 | FT2a (31114633) | SNP_31642505 | 10.1 | 33.0 | 3.6 |
| FR-LD (20h) | *qDTF-10* | 10 | E2 (45310798) | SNP_46678320 | 2.0 [ns] | 5.7 | 1.6 |
| | *qDTF-16-1* | 16 | SNP_4900158 | SNP_5373087 | 7.6 | 36.5 | 3.6 |
| | *qDTF-16-2* | 16 | FT2a (31114633) | SNP_31642505 | 3.7 | 16.4 | 2.3 |

Additive effect for the allele from AK16 (days)

Genomic positions for gene-specific markers are presented within parentheses

PVE (%); Percent of variation explained

[ns]; not significant at 5% level based on the permutation test.

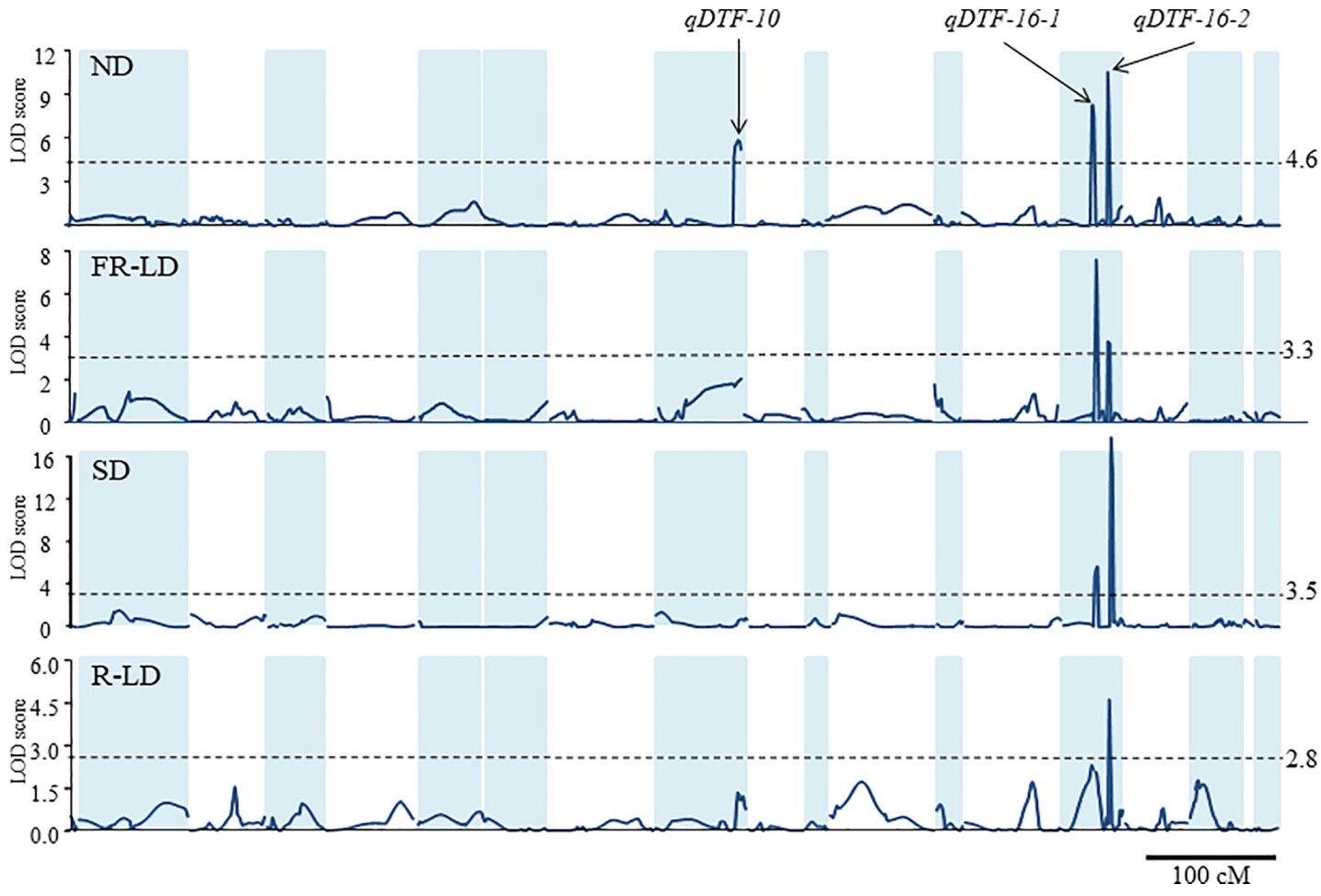

**Fig 4. LOD scores for QTLs controlling the flowering time under four photo-thermal conditions in the F$_6$ progeny of the cross between soybean lines AGS292 and AK16.** Dotted lines indicate significant LOD scores at the 5% level by the 1,000 permutation test.

## Sequence polymorphisms in *FT2a* and *FT5a*

Sequence polymorphisms have been analyzed in the promoters and genic regions of *FT2a* and *FT5a* for various soybean cultivars [8, 9, 26, 28, 49]. Because AK16 possessed the alleles from K3 at both the *FT2a* and *FT5a* loci, we compared these sequences in AGS292 and K3 to identify QTL candidates. The resequencing analysis detected diverse sequence variations, including indels of 10 bp or more, between the two cultivars for *FT2a* (S3 Table). The *FT2a* coding sequence of AGS292 was identical to that of Williams 82 (soybean reference genome sequence), but differed by a nonsynonymous substitution in exon 4 from the coding sequence of K3. The glycine in AGS292 and Williams 82 at the 169[th] aa residue was converted to aspartic acid in K3 (Fig 5). AGS292 and K3 also showed a number of DNA polymorphisms, including four SNPs and four indels in the 2-kb promoter region, one indel in the 5′ UTR, 24 SNPs and 6 indels in the introns, and 1 SNP in the 3′ UTR (Fig 5; S3 Table). In contrast, the *FT5a* sequence was identical between K3 and Williams 82, and it differed by only a SNP in the 3′ UTR from that of AGS292. The SNP generated a MYCCONSENSUSAT cis-element (CAGCTG) in K3 and Williams 82, but not in AGS292 (Fig 5).

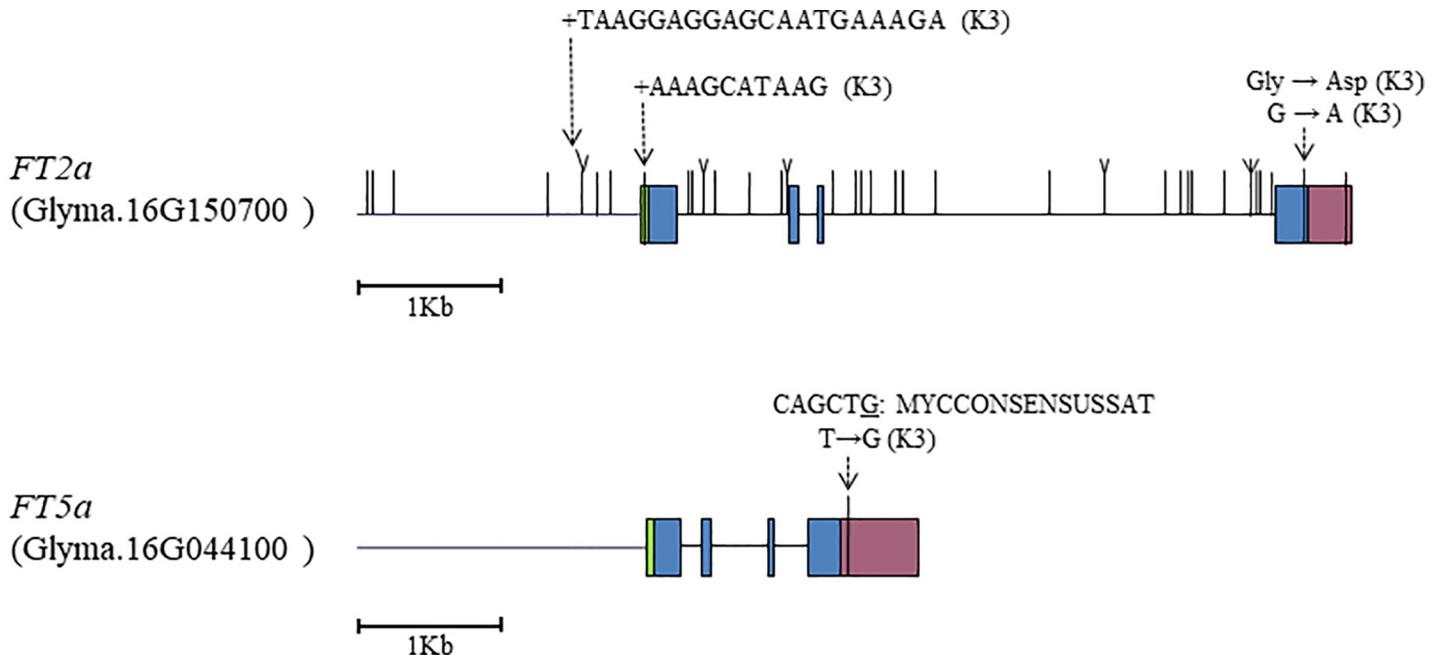

**Fig 5. DNA polymorphisms in *FT2a* and *FT5a* between soybean lines AGS292 and K3.** AK16 possessed the alleles from K3 at both the *FT2a* and *FT5a* loci. Lines indicate SNPs and indels of 1 to 9 bp. Detailed DNA polymorphisms are provided in S3 Table. Green, blue and purple boxes indicate 5′ UTR, exon and 3′ UTR, respectively.

## Transcript profiles for *FT2a*, *FT5a* and *E2*

Transcript abundances of *FT2a* and *FT5a* are closely related to the earliness of flowering under various environmental conditions [12, 26–28]. Expression profiles of *FT2a* and *FT5a* under the SD and R-LD conditions were examined at three time points, 3-, 12- and 21-h Zeitgeber times, in the second and fourth trifoliate-leaf stages (Fig 6A and 6B). In both stages, the transcript abundances of *FT2a* and *FT5a* were almost the same between AGS292 and AK16 under SD conditions, although the expression level of *FT2a* was slightly greater in the former than in the latter. In contrast, the *FT2a* and *FT5a* expression levels were strongly downregulated in AK16, compared with AGS292, under R-LD conditions (Fig 6).

Transcript abundances also varied with thermal conditions (Fig 7A and 7B). At 25°C, the expression levels of *FT5a* were almost the same in AGS292 and AK16, but *FT2a* was slightly upregulated in AK16 compared with in AGS292. At 18°C, the expression levels of *FT2a* and *FT5a* were similar to those at 25°C in AGS292, but they were down-regulated in AK16. The down-regulation was greater in the fourth trifoliate-leaf stage. Intriguingly, the expression levels of *FT2a* and *FT5a* were lower at 32°C, compared with at 25°C, and this hot temperature-related suppression was greater in AK16.

We also examined the transcript profile of *E2* as a possible candidate for *qDTF-10* (Fig 8A and 8B). The transcript abundance of *E2* peaked at 12 ZT and was repressed at 3 and 21 ZT under all conditions. AK16, which possessed the functional *E2* allele, exhibited higher expression levels than AGS292, which possessed the dysfunctional *e2* allele under all of the conditions except for the R-LD at the second trifoliate-leaf stage. The *E2* expression was upregulated at both growing stages in R-LD and at the fourth trifoliate-leaf stage under 32°C conditions, compared with SD and the other two thermal conditions (18°C and 25°C), respectively. However, there was no marked difference in the responses to different photoperiods and thermal conditions between AGS292 and AK16.

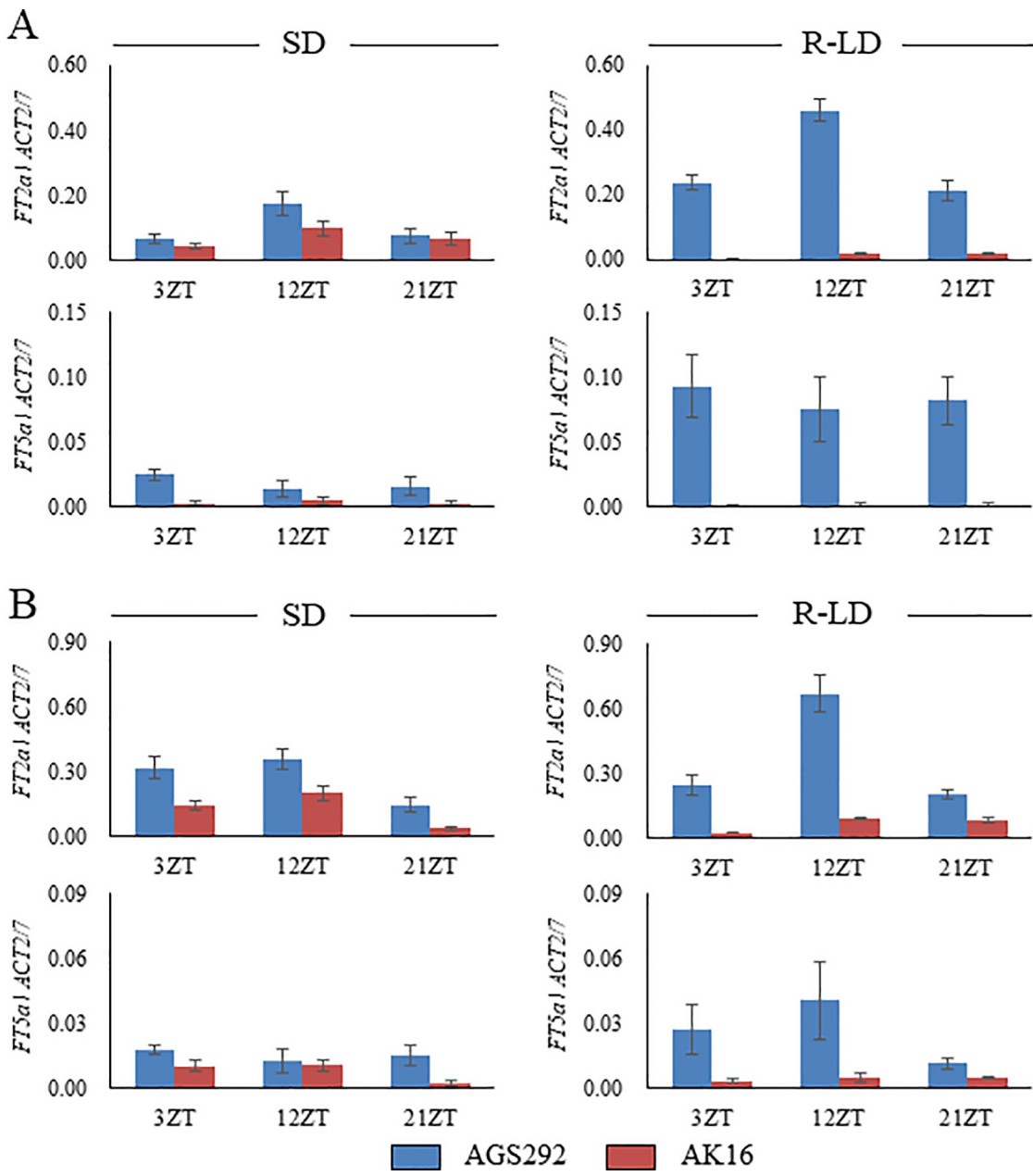

**Fig 6. Expression profiles of *FT2a* and *FT5a* in soybean lines AGS292 and AK16 under SD and R-LD conditions.** (A) Second trifoliate-leaf stage, (B) Fourth trifoliate-leaf stage.

## Discussion

### Characteristics of the LF habit from K3 in a photoperiod-insensitive genetic background

LJ in soybean has been characterized using reciprocal transfer experiments from inductive SD to non-inductive LD conditions or vice versa. The juvenile phase varies from ~10 to 30 d among soybean cultivars [29, 50–53]. However, Cober [54] demonstrated that the LJ cultivars Paranagoiana (*e6/e6*) and PI159925 (*j/j*) retained the sensitivity to shorter photoperiods of less

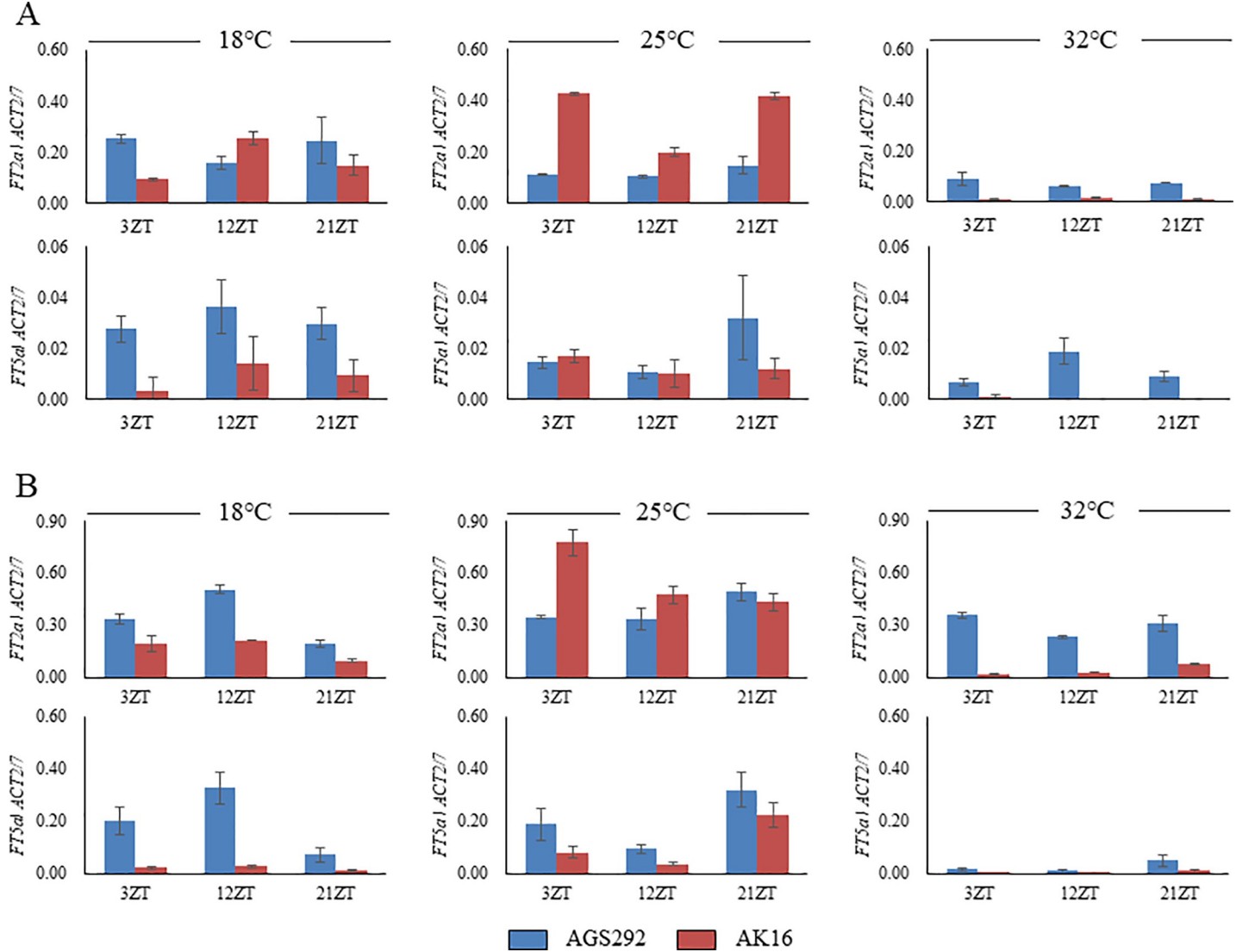

**Fig 7. Expression profiles of *FT2a* and *FT5a* in soybean lines AGS292 and AK16 under different thermal conditions.** (A) Second trifoliate-leaf stage, (B) Fourth trifoliate-leaf stage.

than 12 h, while the conventional juvenile genotype lacking the juvenile phase did not. He assumed that the juvenile phases of these LJ cultivars lasted for a maximum of 5 d, and a flowering delay of more than 5 d was likely caused by photoperiod responses. AK16 used in this study did not possess the *j* allele, but exhibited a flowering delay of 5 to 7 d with 8-h to 16-h photoperiods at a constant 25°C relative to the flowering time of AGS292. The delay increased to 11.7 d with R-enriched LDs of 20 h and to 20.1 d with a 16-h photoperiod at a constant 32°C. Thus, the LF habit introduced from K3 into the photoperiod-insensitive genetic background may involve a basic difference in the activity of floral induction itself and the suppression of flowering by higher temperatures and longer day lengths generated by R-enriched light sources.

AK16 has the double-recessive genotype (*e3/e4*), which enables flowering under FR-enriched LD conditions [55–57]. It did not respond to FR-LD conditions, like AGS292, and the flowering time was almost the same under ND and FR-LD conditions (Fig 2). However,

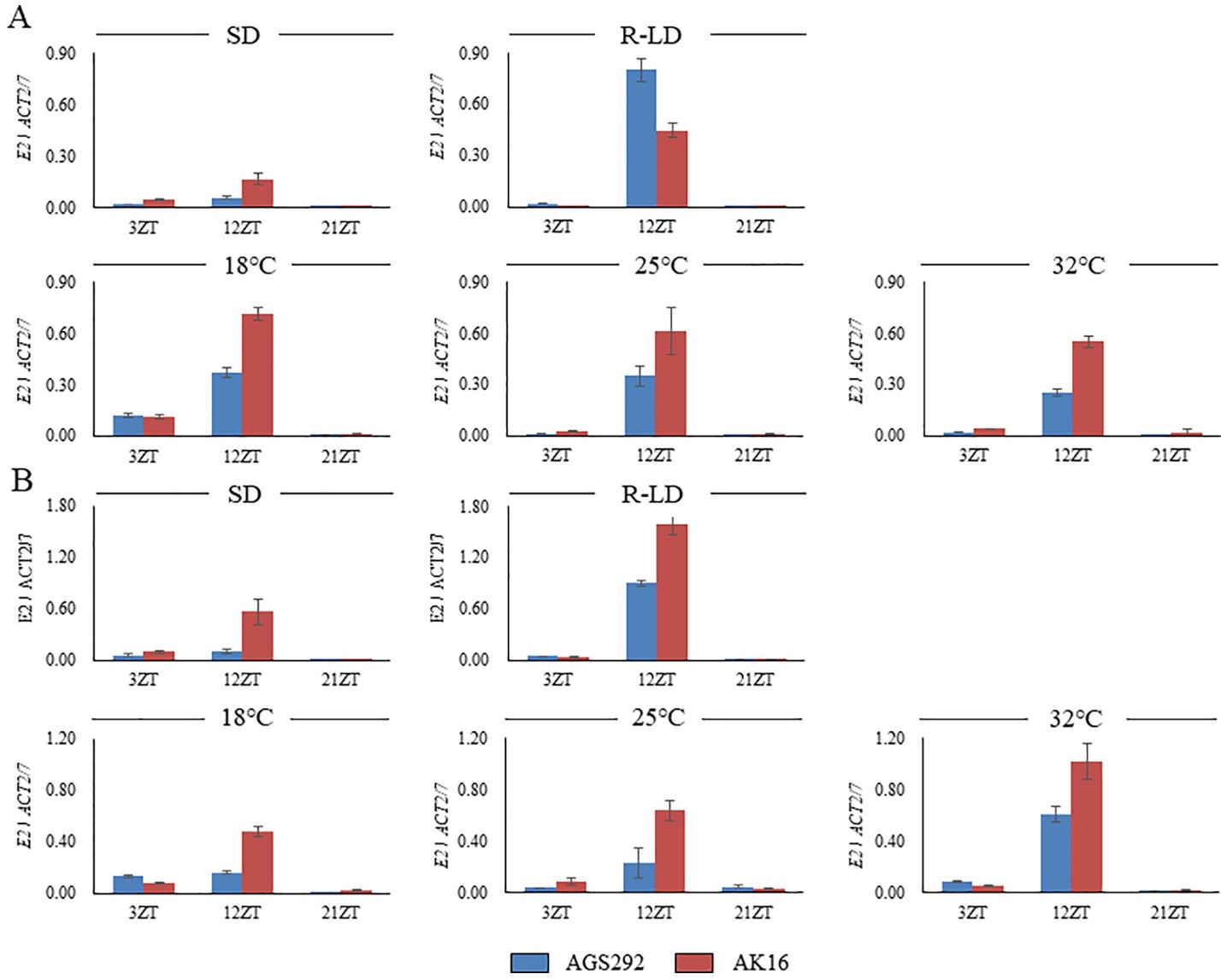

**Fig 8. Expression profiles of *E2* in soybean lines AGS292 and AK16 under different photo-thermal conditions.** (A) Second trifoliate-leaf stage, (B) Fourth trifoliate-leaf stage.

AK16 retained a sensitivity to photoperiods supplied by R-enriched lights (Fig 1A). This was an unexpected result because the response of flowering to R-LD is controlled only by the *E3* gene [58, 59]. *E3* and *E4* participate in different functions of PHYA in Arabidopsis [11, 15]. E4 and its homolog GmPHYA1 are redundantly responsible for the de-etiolation response of hypocotyls to FR light, and in an *e3/e3* genotype, *e4/e4* controls flowering under FR-LD conditions [1, 11, 55–57]. In contrast, *E3* is responsible for PHYA's photoreceptor function for R-light with a high photon irradiance level [15]. The QTL mapping in the original RIL population from a cross between AGS292 and K3 segregating at the *E3* locus demonstrated that *E3* was involved in the control of flowering under a wide range of photoperiods, from the SD conditions of low latitudes to the LD conditions of high latitudes [22]. Because AK16 lacks the functional E3 protein, the residual response to the R-LD conditions might be attributed to the

other PHYA proteins, such as GmPHYA1 and GmPHYA4 [11, 12], other phytochromes, such as PHYB and PHYE [60], and/or the blue light photoreceptor CRYPTOCHROME 2 [61, 62].

In AK16, flowering was inhibited by hot temperatures (Fig 1B). Compared with constant 25˚C, constant or daytime 32˚C promoted flowering in AGS292, but inhibited it in AK16. As in other annual plant species [63–65], increasing the temperature accelerates flowering in soybean [29]. In contrast to the photoperiodic regulation, however, little is known about the genetic variability in the thermoregulation of flowering in soybean. Cober et al. [66] found that different temperatures affected flowering under LD conditions of 16 h or longer. Compared with low temperature (18˚C), a hot temperature (28˚C) markedly delayed the flowering, and the effect was more pronounced in late maturing photoperiod-sensitive genotypes carrying two or more dominant alleles at the *E1*, *E3*, *E4* and *E7* loci relative to the photoperiod-insensitive lines. Whether the inhibition of flowering by hot temperatures observed in AK16 is a common characteristic of the LF habit of LJ cultivars at low latitudes should be determined in a further study. AK16 could be used as a plant resource in the molecular and genetic dissection of the thermoregulation of flowering in soybean.

### QTLs for the LF habit in the photoperiod-insensitive genetic background

In this study, we identified three QTLs (*qDTF-10*, *qDTF-16-1* and *qDTF-16-2*) controlling the LF habit that originated from the Thai cultivar K3 in a photoperiod-insensitive genetic background (Table 1 and Fig 4). *qDTF-10*, *qDTF-16-1* and *qDTF-16-2* were co-localized with gene-specific DNA markers for *E2*, *FT5a* and *FT2a*, respectively (Table 1). The intervals between the left and right markers of respective QTLs contained from 30 annotated genes in *qDTF-16-1* to 167 annotated genes in *qDTF-10* annotated genes including several orthologues of *Arabidopsis* flowering genes (S2 Table). However, these three genes may be probable candidates responsible for the QTLs, because their allelic effects on flowering have been revealed in diverse genetic backgrounds [8, 9, 20, 22, 26, 28, 67].

Of the three QTLs, the *qDTF-10* (*E2*) was detected only under outdoor ND and FR-LD conditions, whereas the other two were detected in all the environments, although the effect at *qDTF-16-1* was not significant at the 5% level under R-LD conditions as assessed by genome-wide permutation tests. In the original RIL population of the cross between AGS292 and K3 segregating for *E3* and *E4*, four QTLs corresponding to *FT2a*, *FT5a*, *E2* and *E3* were detected in the four SD environments tested [22]. The effects of these QTLs, except for the QTL corresponding to *E3*, which was not targeted in this study, could be confirmed in the photoperiod-insensitive genetic background as well.

*FT2a* is an important floral inducer in soybean [13]. When overexpressed using the Cauliflower mosaic virus 35S promoter or knocked out using CRISPR-Cas9, *FT2a* promotes and inhibits flowering, respectively [68–70]. Furthermore, the DNA polymorphisms in the *FT2a* genomic region were associated with variation in flowering time in soybean germplasm having diverse flowering habits. In particular, the non-synonymous nucleotide substitution at exon 4 from guanine to adenine, which converts the glycine to the aspartic acid at the 169[th] aa residue, was highly associated with the late-flowering habit in the world-mini soybean core collection [9] and 127 varieties evenly covering diverse maturity groups [8]. The resequencing analyses detected diverse DNA polymorphisms between AGS292 and K3, including the indels of 20 bp and 10 bp in the promoter and 5′ UTR, respectively, as well as the non-synonymous SNP at exon 4, in which K3 possessed adenine (Fig 5), as in the late-flowering cultivars tested by Jiang et al. [8] and Ogiso-Tanaka et al. [9]. The aa substitutions in the external loop of the FT protein convert its function from a floral activator to a repressor [71, 72]. The glycine at the 169[th] aa residue of FT2a in AGS292 and Williams 82 is highly conserved in FT-like phosphatidylethanolamine-binding

proteins of diverse plant species, whereas the aspartic acid at the position observed in K3 is a characteristic of the floral repressor TFL1-like proteins [73]. Thus, the conversion from glycine to aspartic acid at the 169[th] aa residue may result in a loss of FT2a's function as a floral inducer. Thus, the variant FT2a protein of K3 is most likely involved in the delay of flowering under SD conditions as shown in the original RIL population [22] and also under various photoperiod conditions in a photoperiod-insensitive genetic background in this study. The effect of the aa substitution from glycine to the aspartic acid on floral induction should be validated using ectopic expression analyses in a further study. In addition to this missense variant, another missense and one flame-shift variants were also detected in the accessions introduced from lower latitudes, such as in Taiwan, and South-eastern and South Asian countries [9]. The dysfunctional variants of FT2a might, therefore, have been selected for as LJ genes during the adaptation of soybeans to lower latitudinal environments.

*FT5a*, another important *FT* ortholog in soybean, could be considered a candidate for *qDTF-16-1*. However, the DNA markers with the largest LOD scores varied with the environments tested. The intervals with the largest LOD scores included the gene-specific marker for *FT5a* under all the conditions, except FR-LD conditions, where the greatest LOD score was located in the interval of the next two SNPs, SNP_4900158 and SNP_5373087, approximately 400 kb from the former (Table 1). The QTL analysis of the original RIL population also detected two adjacent QTLs in the genomic region of *qDTF-16-1* [22]. It should be determined in a further study whether a common QTL or each of the two closely linked QTLs is involved in flowering times under different photo-thermal conditions.

The *FT5a* sequences were almost the same between AGS292 and K3, except for a SNP in the 3′ UTR of the latter, which generated a cis-element, MYCCONSENSUSAT, a binding site for the basic-helix-loop-helix transcription factor MYC2 (Fig 5). The MYC2 protein in *Arabidopsis* is a master regulator in various jasmonate-regulated physiological and developmental pathways, such as lateral and adventitious root formation, flowering time and shade avoidance syndrome [74]. MYC2, together with MYC3 and MYC4, redundantly regulates flowering by modulating the transcription of *FT* under both LD and SD conditions [75]. Therefore, the MYCCONSENSUSAT element might function as a binding site of soybean MYC2 orthologs to control the *FT5a* expression in response to various environmental stresses. The *FT5a* expression level in AK16 was subject to repression by severe environmental stresses, such as lower or higher temperatures (Figs 6 and 7). A further study is needed to determine the roles of *cis*-elements in *FT5a* expression in responses to environmental stimuli.

## Molecular mechanisms underlying the LJ trait in soybean

*J* encodes the soybean ortholog of *Arabidopsis* EARLY FLOWERING 3, a component of the evening complex [32, 33]. It physically associates with the *E1* promoter to downregulate its transcription, relieving the repression of *FT2a* and *FT5a* for floral induction under SD conditions [33]. The recessive allele *j* is a loss-of-function allele that cannot repress *E1* expression, resulting, in turn, in the downregulation of *FT* expression and the delay of flowering under SD conditions [33]. In addition to the *j* gene, the missense variant of *FT2a* possessed by K3 may also be involved in LJ. The LJ in soybean might, therefore, reflect a slower floral evocation process owing to the depressed function of *FT2a* resulting from missense or nonsense mutations (this study, [9]) or from the repressed transcription of *FT2a* and *FT5a* associated with the upregulation of *E1*, which is released from repression by *J* under SD conditions [33]. Thus, the LJ trait could result partly from a combination of various dysfunctional alleles at *E9* (*FT2a*) and the loci involved in the PHYA-E1 module, which is an important pathway for photoperiodic flowering in soybean [2]. Further molecular dissections of the LF habit will facilitate our

understanding of the molecular mechanisms regulating flowering in soybean and aid in the marker-assisted breeding of soybean cultivars in a wide range of latitudes.

## Supporting information

**S1 Table. Primer sequences for gene-specific DNA markers and expression analyses used in this study.**
(XLSX)

**S2 Table. Annotated genes in the intervals of the left and right markers of QTLs for days to flowering.**
(XLSX)

**S3 Table. DNA polymorphisms in the genomic region of *FT2a* between AGS292 and K3.**
(XLSX)

## Acknowledgments

We thank Drs. Pornpan Poopronpan of Maejo University and Pirassak Srinives of Kasetsart University for providing us the seed of RIL population. We thank Lesley Benyon, PhD, from Edanz Group (www.edanzediting.com/ac) for editing a draft of this manuscript.

## Author Contributions

**Conceptualization:** Jun Abe.

**Data curation:** Fei Sun, Meilan Xu, Atsushi J. Nagano, Satoshi Watanabe, Fanjiang Kong, Baohui Liu, Tetsuya Yamada, Jun Abe.

**Formal analysis:** Jun Abe.

**Funding acquisition:** Fanjiang Kong, Baohui Liu, Jun Abe.

**Investigation:** Fei Sun, Meilan Xu, Jianghui Zhu, Tetsuya Yamada.

**Methodology:** Cheolwoo Park, Maria Stefanie Dwiyanti.

**Resources:** Satoshi Watanabe, Fanjiang Kong, Baohui Liu.

**Software:** Cheolwoo Park, Maria Stefanie Dwiyanti.

**Writing – original draft:** Fei Sun, Jun Abe.

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
