## [Decision Letter · Decision Letter 0]

21 Oct 2019

PONE-D-19-22803

Characterization and quantitative trait locus mapping of late-flowering in a Thai soybean cultivar with a photoperiod-insensitive genetic background

PLOS ONE

Dear Dr Abe,

Thank you for submitting your manuscript to PLOS ONE. After careful consideration, we feel that it has merit but does not fully meet PLOS ONE’s publication criteria as it currently stands. Therefore, we invite you to submit a revised version of the manuscript that addresses the points raised during the review process.

We would appreciate receiving your revised manuscript by Dec 05 2019 11:59PM. To enhance the reproducibility of your results, we recommend that if applicable you deposit your laboratory protocols in protocols.io, where a protocol can be assigned its own identifier (DOI) such that it can be cited independently in the future. For instructions see: http://journals.plos.org/plosone/s/submission-guidelines#loc-laboratory-protocols

We look forward to receiving your revised manuscript.

Kind regards,

Maoteng Li

Academic Editor

PLOS ONE

**Journal Requirements:**

**Comments to the Author**

1. Is the manuscript technically sound, and do the data support the conclusions?

Reviewer #1: Yes

2. Has the statistical analysis been performed appropriately and rigorously? 

Reviewer #1: Yes

3. Have the authors made all data underlying the findings in their manuscript fully available?

Reviewer #1: Yes

4. Is the manuscript presented in an intelligible fashion and written in standard English?

Reviewer #1: Yes

5. Review Comments to the Author

Reviewer #1: The study ‘Characterization and quantitative trait locus mapping of late flowering in a Thai soybean cultivar with a photoperiod insensitive background’ have significance towards fine tuning flowering, maturity and grain yield of soybean under short day conditions.

Following are Comments:

1. Please provide genomic positions of gene specific markers in table 1?

2. No information provided on the genes predicted between left marker and right marker confidence intervals of identified QTLs. Are there only single gene present in C.I. of each of identified QTL?

3. Please correct ‘IcMAPPING’ to ‘IciMapping’ in Line no. 321 of Page no. 14

4. Transcript profiles of FT2a and FT5a were analyzed but why not E2? Although E2 was not identified in all four environments, transcript profiles in SD and LD environments may have supported it.

6. PLOS authors have the option to publish the peer review history of their article (what does this mean?). If published, this will include your full peer review and any attached files.

Reviewer #1: No

---

## [Author Response · Author response to Decision Letter 0]

15 Nov 2019

We appreciate the reviewer’s comments and suggestions. We revised the points indicated as follows:

1. Please provide genomic positions of gene specific markers in table 1?

(Response) We added the genomic positions of SNPs targeted by gene-specific markers to Table 1, and added the sentence ‘Genomic positions for gene-specific markers are presented within parentheses’ as a footnote. 

2. No information provided on the genes predicted between left marker and right marker confidence intervals of identified QTLs. Are there only single gene present in C.I. of each of identified QTL?

(Response) We have provided information on the annotated genes in the intervals between the left and right markers in a new supplemental table, ‘S2 Table’ (thereby, the original S2 Table was changed to S3 Table in the revised manuscript), and described the numbers of annotated genes in the intervals in the ‘Segregation of flowering time and QTL analysis’ subsection of the Results section in the revised manuscript, as follows:

“The intervals between the left and right markers contained 30, 50 and 167 annotated genes for qDTF-16-1, qDTF-16-2 and qDTF-10, respectively, in the reference genome Williams 82.v2 (S2 Table).”

In addition, we corrected the original sentence “The co-localization with gene-specific DNA markers suggested that qDTF-10, qDTF-16-1 and qDTF-16-2 most likely corresponded to E2, FT5a and FT2a, respectively (Table 1).” in the ‘QTLs for the LF habit in the photoperiod-insensitive genetic background’ subsection of the Discussion section as follows:

“qDTF-10, qDTF-16-1 and qDTF-16-2 were co-localized with gene-specific DNA markers for E2, FT5a and FT2a, respectively (Table 1). The intervals between the left and right markers of respective QTLs contained from 30 annotated genes in qDTF-16-1 to 167 annotated genes in qDTF-10 annotated genes including several orthologues of Arabidopsis flowering genes (S2 Table). However, these three genes may be probable candidates responsible for the QTLs, because their allelic effects on flowering have been revealed in diverse genetic backgrounds [8, 9, 20, 22, 26, 28, 67].” Here we added a new reference as reference number 67 (Wang Y, Gu Y, Gao H, Qiu L, Chang R, Chen S, et al. Molecular and geographic evolutionary support for the essential role of GIGANTEAa in soybean domestication of flowering time. BMC Evol Biol. 2016; 16: 79). We added this citation to the list of the references.

3. Please correct ‘IcMAPPING’ to ‘IciMapping’ in Line no. 321 of Page no. 14

(Response) We corrected our mistake as indicated by the reviewer.

4. Transcript profiles of FT2a and FT5a were analyzed but why not E2? Although E2 was not identified in all four environments, transcript profiles in SD and LD environments may have supported it.

(Response) Thank you for your comment. We added the results of the E2 expression analysis to the revised manuscript. Accordingly we added information on E2 to the ‘Expression analysis’ subsection of the Materials and Methods section and to the ‘Transcript profiles for FT2a, FT5a and E2’ subsection of the Results section. The primers were also added to S1 table. We made a new figure (Figure 8) based on the results of the expression analysis, and described the results in the ‘Transcript profiles for FT2a, FT5a and E2’ subsection of the Results section in the revised manuscript, as follows:

“We also examined the transcript profile of E2 as a possible candidate for qDTF-10 (Fig 8A–B). The transcript abundance of E2 peaked at 12 ZT and was repressed at 3 and 21 ZT under all conditions. AK16, which possessed the functional E2 allele, exhibited higher expression levels than AGS292, which possessed the dysfunctional e2 allele, under all conditions, except for the R-LD at the second trifoliate-leaf stage. The E2 expression was upregulated at both growing stages in R-LD and at the fourth trifoliate-leaf stage under 32°C conditions, compared with SD and the other two thermal conditions (18°C and 25°C), respectively. However, there was no marked difference in the responses to different photoperiods and thermal conditions between AGS292 and AK16.”

In addition to the corrections in response to the reviewer’s comments, we made five corrections in the revised manuscript, as follows:

1. [TITLE] We would like to change the title of the paper from the original ‘Characterization and quantitative trait locus mapping of late-flowering in a Thai soybean cultivar with a photoperiod-insensitive genetic background’ to ‘Characterization and quantitative trait locus mapping of late-flowering from a Thai soybean cultivar introduced into a photoperiod insensitive genetic background’ in the revised one (the corrected portions were underlined). We believe the original title may lead the reader to believe that the Thai cultivar itself possesses a photoperiod-insensitive genetic background. 

2. [Discussion, lines 515 to 518] We mad changes to the following original sentences: “The intervals with the largest LOD scores included the gene-specific marker for FT5a under all the conditions, except R-LD conditions, where the greatest LOD score was located in the interval of the next two SNPs, SNP 4900158 and SNP….”. We corrected the term “R-LD” to “FR-LD” and added the appropriate identifying number after SNP, as follows (the corrected portions are underlined): “The interval with the largest LOD scores varied with the environments tested. The intervals with the largest LOD scores included the gene-specific marker for FT5a under all the conditions, except FR-LD conditions, where the greatest LOD score was located in the interval of the next two SNPs, SNP 4900158 and SNP 53735087…”. 

3. [Discussion, line 547] We added the citations ([this study, 9]) after “nonsense mutations”, as follows (the corrected portions are underlined): “The LJ in soybean might, therefore, reflect a slower floral evocation process owing to the depressed function of FT2a resulting from missense or nonsense mutations [this study, 9] or from …”.

4. [Figure 4] We corrected the order of SNP markers in Chromosome 10, where the QTL for flowering (corresponding to E2) was located, because the SNPs in this chromosome in the original Figure 4 was ordered in the opposite direction of their genomic positions in the Williams 82 reference genome. In the revised manuscript, we presented the corrected figure with the SNP markers in the correct order relative to the reference genome.

5. [Table 1] We corrected the QTL name from qFT to qDTF. In addition, the SNPs for the left and right markers at qDTF-10 in ND and FR-LD were placed in the correct order relative to their genomic positions in the Williams 82 reference genome. 

We also corrected wrong citations as follows:

1. Line 69 (Introduction); We corrected 9 to 10.

2. Line 449 (Discussion); We corrected 61–63 to 63–65.

3. Line 452 (Discussion); We corrected 54 to 66. Reference 54 was the incorrect citation. The proper reference is ‘Cober ER, Stewart DW, Voldeng HD. Photoperiod and temperature responses in early-maturing near-isogenic soybean lines. Crop Sci. 2001; 41: 721–727’. We added this citation to the list of the references. The original reference numbers of 66-73 were thus changed to the reference number of 68-75 in the revised manuscript, by adding two references (66 and 67).

4. Line 484 (Discussion); We corrected 12 to 13.

---

## [Editor Report · Decision Letter 1]

20 Nov 2019

Characterization and quantitative trait locus mapping of late-flowering from a Thai soybean cultivar introduced into a photoperiod-insensitive genetic background

PONE-D-19-22803R1

Dear Dr. Abe,

We are pleased to inform you that your manuscript has been judged scientifically suitable for publication and will be formally accepted for publication once it complies with all outstanding technical requirements.

With kind regards,

Maoteng Li

Academic Editor

PLOS ONE
---

## [Editor Report · Acceptance letter]

26 Nov 2019

PONE-D-19-22803R1 

Characterization and quantitative trait locus mapping of late-flowering from a Thai soybean cultivar introduced into a photoperiod-insensitive genetic background 

Dear Dr. Abe:

I am pleased to inform you that your manuscript has been deemed suitable for publication in PLOS ONE. Congratulations! Your manuscript is now with our production department. 

With kind regards,

on behalf of

Dr. Maoteng Li 

Academic Editor

PLOS ONE